# Fingerprint Oxygen Redox Reactions in Batteries through High-Efficiency Mapping of Resonant Inelastic X-ray Scattering

**Jinpeng Wu [1,2], Qinghao Li [2,3], Shawn Sallis [2,4], Zengqing Zhuo [2,5], William E. Gent [1,2] , William C. Chueh [1], Shishen Yan [3], Yi-de Chuang [2] and Wanli Yang [2,* ]**

[1] Stanford Institute for Materials and Energy Sciences, Stanford University, Stanford, CA 94305, USA; jinpeng@stanford.edu (J.W.); weg44@stanford.edu (W.E.G.); wchueh@stanford.edu (W.C.C.)

[2] Advanced Light Source, Lawrence Berkeley National Laboratory, 1 Cyclotron Road, Berkeley, CA 94720, USA; liqinghaosdu@163.com (Q.L.); ssallis@lbl.gov (S.S.); zhuozq1990@gmail.com (Z.Z.); YChuang@lbl.gov (Y.-d.C.)

[3] School of Physics, National Key Laboratory of Crystal Materials, Shandong University, Jinan 250100, China; shishenyan@sdu.edu.cn

[4] Department of Physics, Applied Physics and Astronomy, Binghamton University, Binghamton, NY 13902, USA

[5] School of Advanced Materials, Peking University Shenzhen Graduate School, Shenzhen 518055, China

\* Correspondence: wlyang@lbl.gov; Tel.: +1-510-495-4989

**Abstract:** Realizing reversible reduction-oxidation (redox) reactions of lattice oxygen in batteries is a promising way to improve the energy and power density. However, conventional oxygen absorption spectroscopy fails to distinguish the critical oxygen chemistry in oxide-based battery electrodes. Therefore, high-efficiency full-range mapping of resonant inelastic X-ray scattering (mRIXS) has been developed as a reliable probe of oxygen redox reactions. Here, based on mRIXS results collected from a series of $Li_{1.17}Ni_{0.21}Co_{0.08}Mn_{0.54}O_2$ electrodes at different electrochemical states and its comparison with peroxides, we provide a comprehensive analysis of five components observed in the mRIXS results. While all the five components evolve upon electrochemical cycling, only two of them correspond to the critical states associated with oxygen redox reactions. One is a specific feature at 531.0 eV excitation and 523.7 eV emission energy, the other is a low-energy loss feature. We show that both features evolve with electrochemical cycling of $Li_{1.17}Ni_{0.21}Co_{0.08}Mn_{0.54}O_2$ electrodes, and could be used for characterizing oxidized oxygen states in the lattice of battery electrodes. This work provides an important benchmark for a complete assignment of all mRIXS features collected from battery materials, which sets a general foundation for future studies in characterization, analysis, and theoretical calculation for probing and understanding oxygen redox reactions.

**Keywords:** oxygen redox; battery electrode; layered oxide; Li-ion battery; resonant inelastic X-ray scattering

---

## 1. Introduction

Developing high energy-density and low-cost energy storage devices has become crucial for effectively addressing energy and environmental challenges, with a projected market expansion for 10 times in this decade [1]. While the Li-ion battery system has been a ubiquitous energy storage solution in modern electronics and electrical vehicles due to its high energy and power density [2], the Na-ion battery has become a promising solution for grid-scale applications by virtue of its abundant resource and low cost [3]. However, improving battery technology to meet the requirements of modern

energy storage remains a formidable challenge, which calls for conceptual breakthroughs and advanced characterizations to explore new solutions beyond conventional systems.

As a bottleneck on the energy density of a battery, the cathode (positive electrode) has attracted tremendous interest and great efforts for enhancing its energy and power density [4–6]. Currently, all commercial Li-ion battery cathodes are based on *3d* transition metal (TM) oxide compounds. In a conventional system, only transition-metal (TM) redox is involved in electrochemical cycling, and introducing oxygen (O) redox was believed to be detrimental to the reversibility and safety of a battery, mainly because of the irreversible $O_2$ release and parasitic surface reactions. However, some recent studies suggest that oxygen redox could be reversible and could greatly improve the energy density of battery cathodes [7]. This opens opportunities for developing and optimizing TM oxide materials towards a high energy density beyond conventional TM redox systems.

The practicability of the oxygen redox concept strongly depends on the reversibility of the reaction upon extended electrochemical cycles. Therefore, it is critical to detect the changes of the oxygen states in the electrodes upon electrochemical cycling and evaluate how reversible and stable the reactions are. Additionally, to contribute to the capacity, the reversible oxygen redox reaction needs to take place inside the bulk lattice of the electrode materials. A technical approach for probing the lattice oxygen states thus becomes important. At this time, conventional oxygen spectroscopy, especially O *K*-edge (O-*K*) X-ray photoelectron spectroscopy (XPS) and X-ray absorption spectroscopy (XAS) have been extensively employed for detecting, and sometimes quantifying, the oxygen redox reactions; however, they have suffered various technical issues from the shallow probe depth and/or entangled signals for achieving a reliable probe of oxygen redox states [8]. This is because the non-divalent oxygen states, e.g., peroxides and $O_2$ gas, manifest itself in O-*K* XAS within the energy range of the so-called "pre-edge" range at about 528–534 eV [9,10], where strong TM-O hybridization features dominate the XAS signals [11]. The TM-O hybridization strength in electrode systems evolves with electrochemical cycling due to multiple factors, such as the changing oxidation states, the structural evolution, and the overall covalency variation, which leads to a general increase (decrease) of the O-*K* XAS pre-edge area at charged (discharged) states in almost all electrode systems with or without oxygen redox reactions, e.g., in olivine $LiFePO_4$ [12], and Spinel $LiNi_{0.5}Mn_{1.5}O_4$ [13]. Therefore, further resolving of the O-*K* XAS pre-edge features is required to rule out the strong effects from hybridization, and to achieve a direct and reliable probe of the oxygen redox states in battery electrodes.

Such a technical challenge is now solved through ultra-high efficiency mapping of resonant inelastic X-ray scattering (mRIXS) [14,15], which could decipher the different components buried in the XAS pre-edge features and provide a bulk probe of the critical oxygen states in battery electrodes [8]. Furthermore, because mRIXS probes the unconventional oxygen states along both the excitation and emission energies [16], it reveals the full spectroscopic profile of the oxidized oxygen that could be quantified to watch the evolution of oxygen redox reactions upon electrochemical cycling [17]. It has been established at this time that a striking feature at 523.7 eV emission and 531 eV excitation energies emerges at the charged state, which fingerprints the oxidized oxygen that is involved in the oxygen redox reactions in various Li-ion and Na-ion battery electrodes [17–19].

However, while this specific feature in mRIXS is used as the tool-of-choice for studying the novel oxygen states, it is only one small part of the overall mRIXS signals. More importantly, other than this focused mRIXS feature, all other O-*K* mRIXS features change with electrochemical cycling [17–19]. Therefore, an explicit interpretation is important for clarifying the meaning of all the changing O-*K* mRIXS features observed for battery electrodes at different electrochemical states.

In this work, we provide a careful analysis of all the mRIXS features observed in a series of Li-rich $Li_{1.17}Ni_{0.21}Co_{0.08}Mn_{0.54}O_2$ (LR-NMC) electrodes and its comparison with $Li_2O_2$ results. Although a complete understanding of mRIXS results through theoretical calculations remains a grand challenge, the several decades of knowledge accumulated in the physics field and the direct comparison to XAS results allow us to provide a reliable assignment of all the mRIXS features. We show that the mRIXS signals from oxide electrodes could be attributed to five different origins: (1) The TM(*3d*)-O(*2p*)

hybridization, (2) the general $O^{2-}$ bands mixed with TM(*4s*,*4p*)-O(*2p*) hybridization, (3) the d-d excitations within the TM states, (4) low-energy (<1 eV) excitations associated with oxidized oxygen states, and (5) the non-divalent oxygen states from oxidized oxygen in charged states. Both (4) and (5) features evolves upon oxygen redox states changes, providing spectroscopic fingerprints of oxygen redox reactions in battery electrodes. Feature (5) provides a high statistic probe, while feature (4) is also useful for both chemical analysis and fundamental understandings. This work clarifies the origin of the many mRIXS observations, provides a useful benchmark on understanding the mRIXS results, and inspires guidelines for future mRIXS studies of oxygen redox reactions in battery electrodes.

## 2. Results and Discussion

### 2.1. Materials and O-K mRIXS Technique

The LR-NMC material was synthesized and electrochemically cycled as in the previous report [18], which is briefly described in the Materials and Methods section for readers' convenience. $Li_2O_2$ was purchased from Sigma Aldrich. The electrode displays the typical high voltage plateau for LR-NMC materials during the first charging, which corresponds to a large amount of oxygen redox in the LR-NMC cathode (Figure 1a). Six representative samples were selected at different electrochemical potentials, as indicated by the red dots in Figure 1. Two more electrodes were cycled for 500 cycles to its charged and discharged states with the electrochemical profile reported before [18]. All samples were handled under a high purity Ar environment before being directly coupled to the experimental vacuum chamber to avoid any air exposure. Technically, we noticed the oxidized oxygen signals in some battery electrodes, especially some Li-ion battery materials, will decrease in intensity upon soft X-ray radiation, which behaves the same as $Li_2O_2$ [9]. We therefore carefully tested the radiation dependence of several systems, which will be reported separately. Nonetheless, the radiation sensitivity leads to only underestimated oxidized oxygen signals in sensitive materials, which does not affect the qualitative detection of the emergence of oxygen redox. However, extra caution is necessary for quantitative analysis, which is feasible through a controlled X-ray dose and maintaining an itinerant sample during experiments.

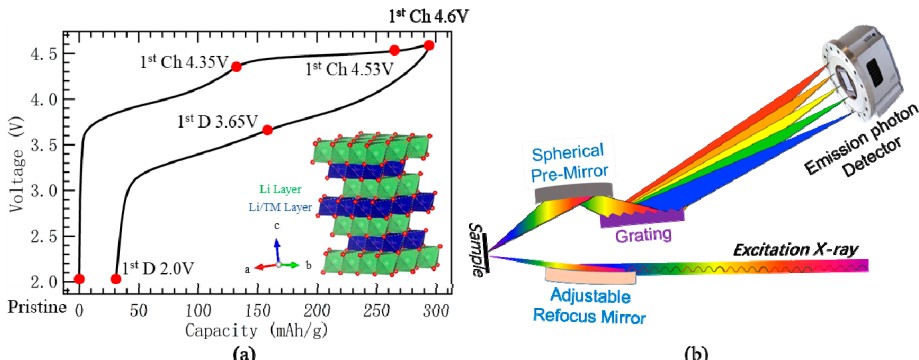

**Figure 1.** (**a**) Electrochemical profile and crystal structure of LR-NMC material. The crystal structure of the LR-NMC material belongs to the *C*2/*m* space group. The high voltage plateau during the first charging cycle corresponds to a large amount of oxygen oxidation. Six representative samples at different electrochemical potentials are marked as the red dots on the electrochemical profile. These electrodes, together with the charged and discharged electrodes after 500 cycles, are selected for mRIXS characterizations. (**b**) Schematic illustration of the mRIXS system. The experimental set-up, other than the light source, consists mainly of a refocus mirror, spectrometer optics with a mirror and a grating, and an X-ray photon detector. Reproduced from Ref. 15 with permission.

mRIXS of this series of LR-NMC electrodes at different electrochemical states were collected in the ultra-high efficient iRIXS endstation at Beamline 8.0.1 of Advanced Light Source [15]. Figure 1b is the schematic illustration of the mRIXS system in the endstation, which has been detailed in previous

work [15]. The incident soft X-ray was generated by the synchrotron light source, concentrated by the bendable mirror, and shed on a sample. The soft X-ray fluorescence photons were emitted from the sample, and were differentiated by an ultra-high efficiency spectrometer equipped with optics and a charge-coupled device (CCD) detector. mRIXS data were collected with a 0.2 eV step size in excitation energy across the full O-*K* absorption edges. At each excitation energy, the collected RIXS signals were normalized to the incident beam flux and data collection time. The normalized RIXS cuts at the individual excitation energy were summarized together into a color-scale mRIXS image with the intensity represented by the color. Details on data processing can be found in a previous publication [14].

## 2.2. Presentations of O-K mRIXS

mRIXS results are 2D images with two energy axes, one corresponding to the indecent X-ray excitation energies, while the other is the emitted photon energies measured through an RIXS spectrometer. Therefore, mRIXS could be presented in three typical ways in the literature (Figure 2), and each would highlight the origin of certain signals from different electronic decays during the spectroscopic process [8].

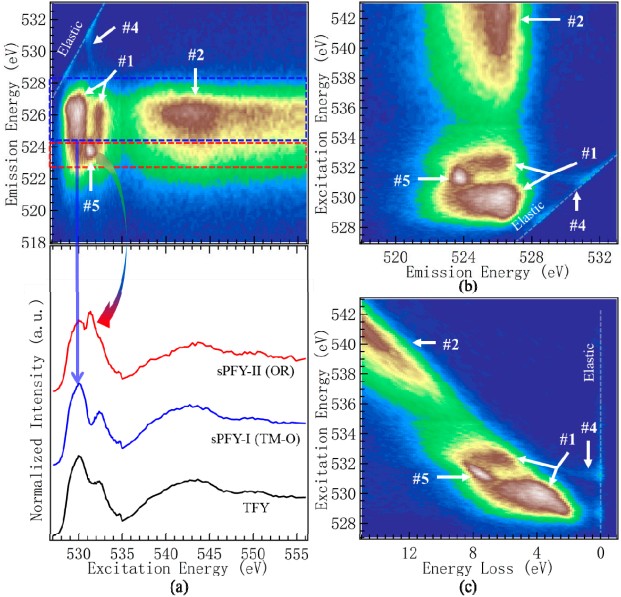

**Figure 2.** O-*K* mRIXS of a fully charged (1st Ch 4.6 V) LR-NMC electrode presented in three different ways to emphasize different features. (**a**) mRIXS image (top) plotted together with extracted XAS channels (bottom). The excitation energy is the horizontal axis, which is preferred by most material scientists because mRIXS results can be compared directly with XAS features. Such direct comparison shows that Feature-1 corresponds to the two main peaks in the O-*K* XAS pre-edge range, which is of TM(3*d*) character from the TM(3*d*)-O(2*p*) hybridization. Feature-2 corresponds to the XAS hump at high energy, and is of TM(4*s*/4*p*) character from TM(4*s*/4*p*)-O(2*p*) hybridization. The striking Feature-5 is the oxidized oxygen feature that could be extracted through sPFY-II by integrating the signals around the characteristic 523.7 eV emission energy range. sPFY-I(TM-O) is the integration of the mRIXS intensity in the emission energy range around 525 eV, emphasizing TM-O hybridization features. It is clear that conventional XAS TFY signals are dominated by TM-O hybridization features, especially in the pre-edge regime. (**b**) A typical way of plotting mRIXS with emission energy along the horizontal axis. The broad Feature 1 and 2 are consistent with typical O-*K* RIXS results of oxides remaining constant around 525 eV emission energy. (**c**) mRIXS results with energy loss values plotted along the horizontal axis. This is preferred in physics because it shows the value of energy loss for specific RIXS excitations, e.g., Feature-4 displays an energy loss of less than 1 eV, which is likely from the excitation of vibronic modes (see text).

First, most material scientists prefer mRIXS images with excitation energies along the horizontal axis. An example is shown in the top of Figure 2a with mRIXS data collected from a fully charged LR-NMC electrode (1st cycle, 4.6 V). The benefit of such plots is the direct correspondence to the conventional XAS spectra upon the same excitation energies (horizontal axis in Figure 2a). Because mRIXS is technically further resolved XAS fluorescence signals along the emission energy (vertical axis in Figure 2a) [8], the vertical integration of the mRIXS intensity corresponds to XAS-type signal channels. Different features could be emphasized depending on the area of integration. This could be seen directly through the comparison between the mRIXS image and the extracted spectra in Figure 2a. For example, the integration of intensities along the whole emission energy range generates the conventional XAS spectrum in the total fluorescence yield (TFY) mode. The O-*K* XAS TFY peaks in the interested 528–534 eV pre-edge range are known to be of TM character from the strong TM($3d$)-O($2p$) hybridization, as explicitly concluded in the seminal study by de Groot et al. in 1989 [11], and later confirmed by extensive theoretical and experimental studies. Indeed, direct comparison between the mRIXS and XAS shows that the pre-edge peaks are from the broad features around the 525 eV emission energy (mRIXS Feature-1), which is also known as the typical (nominally $O^{2-}$) emission energy value for TM oxides [20]. Along the same emission energy range, the broad mRIXS feature above the 535 eV excitation energy (Feature-2) corresponds to the broad hump in XAS at high energies, stemming from the weakly structured TM $4s$ and $4p$ states mixed with oxygen $2p$ bands [11]. The decay from the broad O-$2p$ valence band electrons to the excited core holes leads to the so-called X-ray emission spectroscopy (XES), sometimes called a fluorescence feature in RIXS experiments [8], which dominates the featureless signals above 545 eV excitation energy around the same 525 eV emission energy. A super-partial fluorescence yield (sPFY) by integrating mRIXS intensity only around the 525 eV emission energy (blue frame in Figure 2a) shows that the overall XAS (TFY) line shape is dominated by features that are standard to TM oxides.

Second, the characteristic behavior of an XES type of signal is the constant emission energy defined by the energy difference between the valence-band electrons and the core holes. Figure 2b is a more typical mRIXS plot than Figure 2a, with emission energy as the horizontal axis (Figure 2b). In such a plot, the emission energy of the broad vertical features is clearly shown at 525 eV and are extended to high excitation energies beyond the absorption edges. As explained above, depending on the excitation energy (vertical axis here) range, broad mRIXS features around the 525 eV emission energy are of a TM $3d$ character (Feature-1), TM $4s/p$ character (Feature-2), and XES signals with O-$2p$ character at very high excitation energies.

Third, other than the decays of electrons to the core holes, the excited system after photon absorption triggers various excitations before the core holes are filled, especially when the excitation energy is close to the absorption threshold [8]. Such excitations are critical for understanding the chemistry and physics of a material with specific electronic configurations. For example, for TM $3d$ states, excitations between the occupied and unoccupied $3d$ states manifest themselves in RIXS results with a cost of energy, called "energy loss". Such a *d-d* excitation has typical energy loss values of several eVs, which is far more sensitive to the chemical states than conventional XAS results [21]. More importantly, it has long been found that TM *d-d* excitations could be observed in O-*K* RIXS experiments in many oxide systems with a couple eVs of energy loss [22]. Figure 3 displays the mRIXS images of the pristine (discharged) LR-NMC electrode with the horizontal axis as the emission energy (Figure 3a) and energy loss values (Figure 3b). The *d-d* excitation feature could be seen through the weak intensity parallel to the elastic line. This is also better seen through the individual energy distribution curves at different excitation energies (Figure 3c), where an energy loss of about 2.4 eV could be seen within the excitation energy range for the strongest TM-O feature. Note such a *d-d* excitation feature cannot be observed clearly in charged electrodes, e.g., Figure 2c, which will be discussed below.

Fourth, two associated mRIXS features emerged in the charged electrodes if oxygen redox reactions were involved [8,17–19]. One is a sharp feature at a 523.7 eV emission and 531 eV excitation energy (Feature-5), while the other is a weak feature close to the elastic line at the same excitation

energy (Feature-4). We notice these two features are coupled together in the mRIXS results of both Li-ion and Na-ion battery electrodes, and they both evolve with the electrochemical states if oxygen redox reactions are involved [8,17–19]. The sPFY with signals integrated around the characteristic 523.7 eV emission energy range (red frame in Figure 2a) emphasizes the strong oxygen-redox feature, with a striking peak around 531 eV (sPFY-I in Figure 2a). The associated low-energy excitation feature-4 at the same excitation energy could be evaluated better in the energy-loss plot (Figure 2c), which displays a value of energy loss of about 1 eV. Both of these mRIXS features could be used as signatures of oxidized oxygen states in charged electrodes associated with oxygen redox reactions. The evolution of these features upon electrochemical cycling are elaborated below.

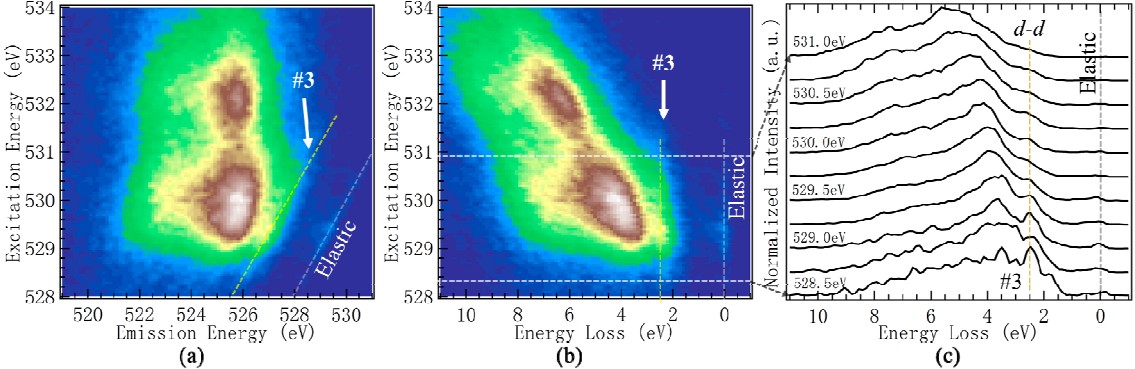

**Figure 3.** O-*K* mRIXS of a pristine LR-NMC electrode in discharged state. (**a**) mRIXS plotted upon emission energy; (**b**) mRIXS plotted upon energy loss; (**c**) individual RIXS cuts within the excitation energy range indicated by the dashed white lines in (**b**). Feature-3 is seen with a 2.4 eV energy loss at energies close to the absorption edge, indicating TM *d-d* excitations in the discharged electrode. Note that such a *d-d* excitation feature is unclear in the charged electrode, as shown in Figures 2 and 4.

## 2.3. mRIXS Features of TM Characters Upon Electrochemical Cycling

Upon electrochemical cycling, the electrode material change in both the structure and electronic states due to the cruel process of alkali ion extraction and insertion. These changes are often associated with each other and play critical roles in understanding the reversibility and stability of battery electrode performance. Indeed, we have found all the five mRIXS features in TM oxide based cathodes evolve with electrochemical operations. Below we first discuss the changes of the features of TM characters (Feature 1–3), then we focus on the features related with the oxygen redox reactions (Feature-4, 5).

The mRIXS feature with sole TM character is the *d-d* excitations with about 2.4 eV energy loss. Such a TM character has long been reported in various oxides with a couple eVs of energy loss [22]. This feature is weak in O-*K* mRIXS results, but is clear in discharged electrodes compared with charged ones (Figure 4). For example, compared with Figure 2a collected from the charged electrode, the *d-d* excitations could be seen clearly in Figure 3 with a 2.4 eV energy loss across the range of 528–531 eV excitation energies. The feature is strongly enhanced with a low excitation energy that is right at the absorption edge (Figure 3). Although weak in general, the *d-d* excitation feature-3 displays a reversible behavior over hundreds of cycles (Figure 4). A complete understanding of these TM *d-d* excitations in O-*K* mRIXS is an interesting topic in fundamental physics that requires complex theoretical calculations. However, the general variation upon electrochemical cycling is likely due to the relatively higher covalence in charged (oxidized) electrodes, which broadens and diminishes such weak features in spectroscopy.

In addition to the change of the TM d-d feature-3 upon cycling, Figure 4 also shows that the broad TM-O hybridization features (Feature-1, 2 in Figure 2) get stronger when the electrodes are charged. The enhancement could be seen throughout the whole initial charging voltage range. Because the oxygen redox reaction takes place only above 5.35 V, such an enhancement of the hybridization

features does not represent oxygen redox activities. As explained in Figure 2a, the enhancement of
TM-O hybridization upon charging is the main reason for the enhanced pre-edge area in the O-*K* XAS
with or without oxygen redox involved. The enhanced intensity could be naturally understood by
considering that the system is oxidized upon electrochemical charging. Oxidations of either TMs or
oxygen led to enhanced hybridization, leading to stronger hybridization features in mRIXS and the
pre-edge in XAS. Note that the other structural effect, especially during the initial charge, may also
play a role for enhancing the TM-O hybridization. Because TM-O hybridization takes place in all oxide
electrodes, the enhancement of hybridization features in mRIXS and XAS exists in all battery electrode
systems, e.g., olivine LiFePO$_4$ [12]. Therefore, although the precise value of the oxygen valence could
be modified in systems with different strengths of hybridization, such an "involvement of oxygen"
through hybridization is fundamentally different from and should not be considered the intrinsic
oxygen redox reactions.

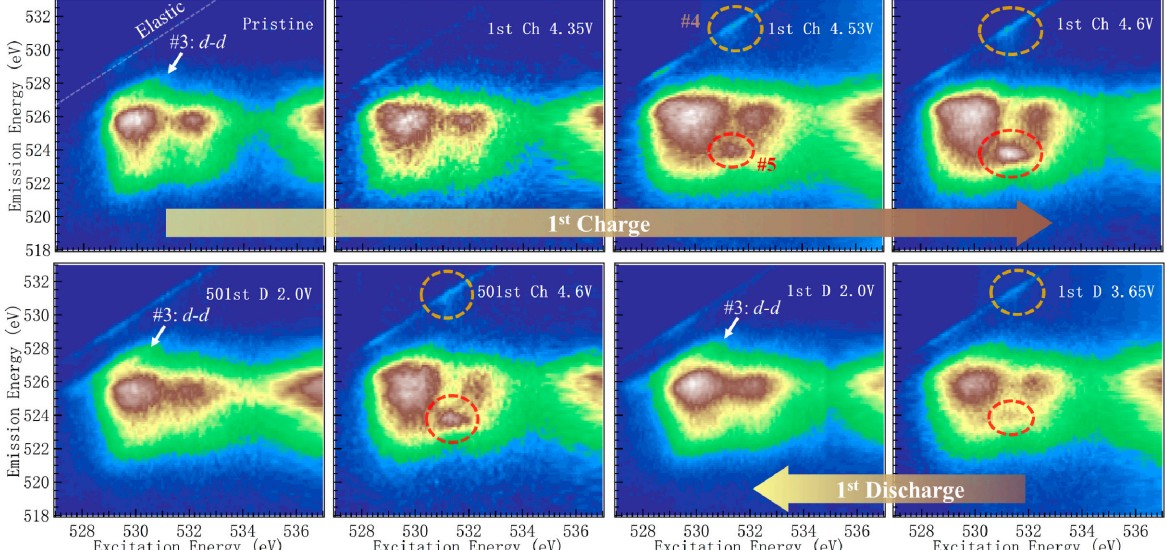

**Figure 4.** mRIXS of a series of LR-NMC electrodes at different electrochemical states indicated
in Figure 1. All five features in mRIXS change upon electrochemical cycling. The broad TM-O
hybridization features are enhanced upon oxidation (charging). The TM *d-d* excitation feature-3 is
clearer in discharged states compared with charged ones. A sharp feature-5 emerges at a 531.0 excitation
energy and 523.7 eV emission energy (red dashed circles) in charged or partially charged electrodes.
Additionally, the intensity of the low energy loss feature-4 (yellow circles) changes upon electrochemical
cycling in the same way as that of the dedicated oxygen-redox feature.

### 2.4. Oxygen-Redox Features in mRIXS

### 2.4.1. Distinct Oxygen Redox Feature

Without the clarification of its fundamental mechanism at this time, oxygen redox reactions
should be dedicated oxidation and reduction reactions of the oxygen in electrode materials. While
the reversibility of oxygen redox reactions is obviously material dependent, a dedicated state of
oxidized oxygen, i.e., non-divalent oxygen, should emerge during the charging and disappear during
discharging for at least one cycle. In such a definition, the release of oxygen from electrode materials
and the associated surface reactions are largely irreversible and should be considered as only an oxygen
oxidation reaction [23–25], not a redox reaction. Therefore, the experimental signature of oxygen redox
reactions requires reversible behaviors of dedicated oxygen signals.

Figure 4 displays that a sharp feature emerges at a 531.0 excitation energy and 523.7 eV emission
energy on the O-*K* mRIXS (Feature-5, red dashed circles) when the initial charging voltage goes beyond
4.35 V. The intensity of the feature is further enhanced during the first charge plateau, reaching its

maximum intensity at the fully charged state at 4.6 V then decreasing during the following discharge process. The emergence and disappearance of this feature during the charge and discharge, respectively, continue for hundreds of cycles, as shown by the electrodes after 500 cycles, providing a reliable signature of the oxygen-redox states in LR-NMC electrodes [18].

### 2.4.2. Low Energy-Loss Feature Associated with Oxygen Redox

Strikingly, the intensity of the low-energy loss feature-4 also evolves with electrochemical cycling. More importantly, as shown in Figure 4 (yellow circles), the intensity of this energy loss feature changes upon electrochemical cycling in exactly the same way as that of the dedicated oxygen-redox feature discussed above. Therefore, the oxidized oxygen in charged electrodes obviously introduces some features with <1 eV energy loss. Although these features are fairly weak compared with feature-5, they sit on a relatively clean background without much overlapping with the broad hybridization feature-1, providing another indicator for detecting the oxidized oxygen. Additionally, low-energy excitations in this energy range are typically from photons or magnons [26]. Observing such a feature in O-*K* implies that the specific vibronic modes are triggered in the oxidized oxygen system.

### 3. Discussion

The analysis of each of the five mRIXS features above indicates that the sharp feature-5 at 523.7 eV emission and 531 eV excitation energies is a reliable fingerprint of the oxygen redox states in the charged LR-NMC electrodes. To compare this feature directly with a reference of the oxidized oxygen state, Figure 5 shows the mRIXS results of both the fully charged LR-NMC electrode and $Li_2O_2$. A vertically extended feature at the same 523.7 eV emission energy is observed in $Li_2O_2$ across the excitation energy range of 529–533 eV, consistent with the broad XAS leading peak [9]. Theoretical calculations have shown that this feature originates from O-*2p* intra-band excitations with electrons excited into unoccupied O-*2p* states that are close to the Fermi level [16]. As *2p* states of $O^{2-}$ are fully occupied, such an excitation is only possible with the existence of unoccupied states in oxidized oxygen, thus providing a spectroscopic signature of oxidized oxygen.

Although Figure 5 confirms that the 523.7 eV emission energy feature-5 is indeed an indicator of oxidized oxygen, the contrast here indicates that the oxidized oxygen states in LR-NMC electrodes is not simply the same as $Li_2O_2$. Mainly, the oxygen redox state in the electrode displays a very sharp mRIXS feature that typically implies a specific excitation in the TM oxide, e.g., a specific charge transfer excitation [27]. However, the oxidized oxygen in $Li_2O_2$ shows a feature much extended along the excitation energy, stemming from intra-band excitations [16]. Intensive theoretical calculations of RIXS are still necessary to fully understand the fundamental origin of the oxygen redox feature in oxide electrodes, which will eventually lead to the ultimate understanding of the fundamental driving force of oxygen redox reactions in battery materials.

The contrast in Figure 5 also shows another important behavior of the oxidized oxygen states: The mRIXS profile along excitation energies strongly depends on the material, which is much sharper in the LR-NMC electrodes compared with that of $Li_2O_2$. We note that such an important profile is largely missing in the conventional RIXS cuts with a coarse step size of excitation energies (Figure 5b,d). Additionally, as described in Figure 4, the intensity of the oxygen-redox feature varies systematically with electrochemical cycling, which could be quantified to monitor the behavior of oxidized oxygen in battery electrodes [17]. Compared with the signal change of a single RIXS cut, quantifications by integrating the full mRIXS profile along both the emission and excitation energies are desirable because of both the improved signal-to-noise ratio and completeness.

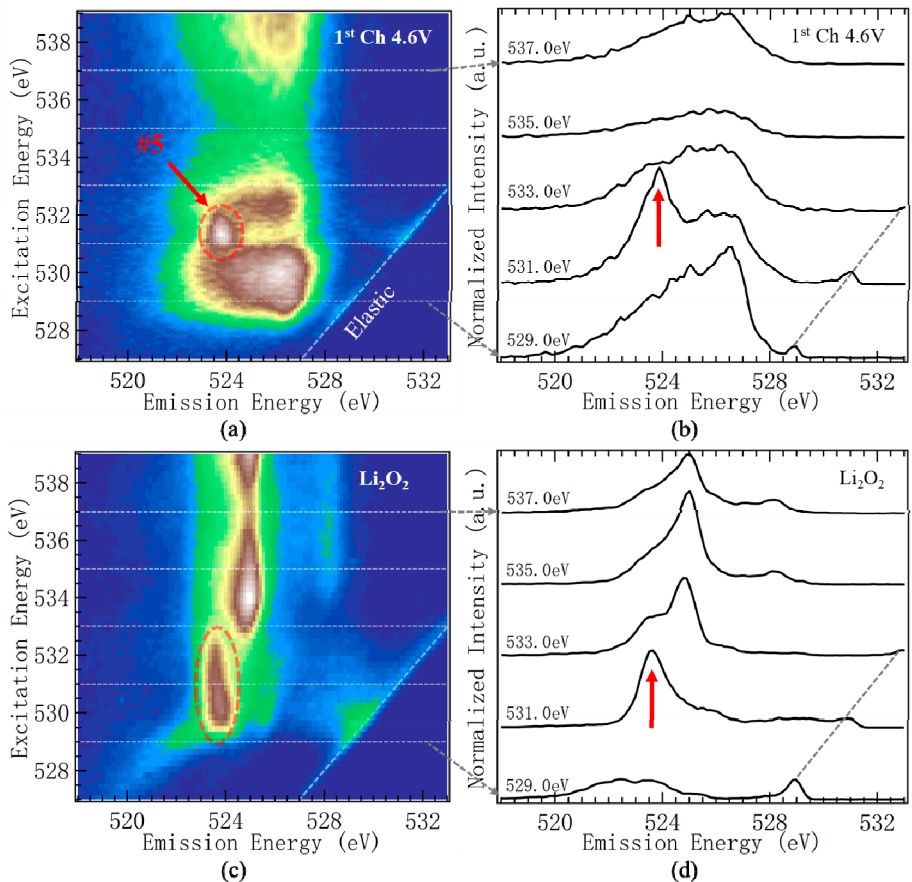

**Figure 5.** Comparison of mRIXS and RIXS cuts between a fully charged LR-NMC electrode and $Li_2O_2$. (**a**) mRIXS of 1st Ch 4.6 V LR-NMC. (**b**) RIXS cuts of 1st Ch 4.6 V LR-NMC with excitation energies noted beside each spectrum. (**c**) mRIXS of $Li_2O_2$; (**d**) RIXS cuts of of $Li_2O_2$. RIXS cuts in the right two panels are collected at five excitation energies with a 2 eV/step from 529 to 537 eV, as indicated by the dashed line in the left panels. While the fully charged LR-NMC shows a much sharper feature along the excitation energy than $Li_2O_2$ on mRIXS, this contrast is mostly missing in RIXS cuts with a large step size in excitation energies.

Additionally, the other oxygen redox indicator, the low energy loss feature-4, should not be depreciated. Technically, this feature, although weak, emerges from a relatively clean signal background away from the strong hybridization and XES features, providing an immaculate signature of oxidized oxygen. Fundamentally, this provides a crucial hint for revealing the mechanism of the oxygen redox states at the molecular level: The configuration of the oxidized oxygen spontaneously triggers excitations with an energy loss of less than 1 eV, likely through coupling to vibronic modes in this energy range.

## 4. Materials and Methods

**Material synthesis.** The pure LR-NMC powder was synthesized via calcining the mixture of the $(Ni_{0.21}Co_{0.08}Mn_{0.54})(OH)_2$ precursor and an appropriate amount of $Li_2CoO_3$ at 900 °C for 10 h. The Li/Ni/Co/Mn molar ratios were 1.434:0.251:0.099:0.650, as measured by inductively coupled plasma mass spectrometry.

**Electrochemical cycling.** 80 wt% LR-NMC was mixed with 10 wt% polyvinylidene fluoride binder (MTI Corporation, Richmond, CA, USA) and 10 wt% carbon black (Timical C65) into *N*-methyl-2-pyrrolidone (Acros Organics, Pittsburgh, PA, USA) and cast onto Al foil. The film after drying was assembled into Coil cell (size CR2016, MTI Corporation) along with two Celgard separators, a Li foil anode (Sigma-Aldrich, Saint Louis, MO, USA), and 1 M LiPF6 in 1:1 (*wt/wt*) ethylene carbonate

(EC)/diethyl carbonate (DEC) electrolyte (Selectilyte LP 40, BASF). The voltage curves were measured at 4 mA/g under a constant temperature of 30 °C.

**mRIXS.** O-*K* mRIXS was collected in ultra-high efficient iRIXS endstation at Beamline 8.0.1 of Advanced Light Source, Lawrence Berkeley National Laboratory [15]. All the samples were prepared in the Ar glove box. After which, the samples were sealed in a specially designed suitcase and transferred into the experimental vacuum chamber. During the whole process, the samples were not exposed to any air. The detailed experimental and data processing protocols are explained in a previous report [14].

## 5. Conclusions

This work provides a comprehensive interpretation of all the five features observed in O-*K* mRIXS of LR-NMC based battery cathodes involving oxygen redox reactions. We have closely studied the evolution of all the mRIXS components upon electrochemical cycling, and found all of them evolve with electrochemical states, but only two of them represents the intrinsic oxygen redox reactions, both of which emerged in the charged electrodes through specific signals from oxidized oxygen. It is concluded that:

1.  The broad mRIXS features around the 525 eV emission energy are of (i) TM-*3d* character at the XAS pre-edge range of 527–534 eV excitation energy (Feature-1) and (ii) TM-*4sp* character at the 537–545 eV excitation energy (Feature-2), in addition to (iii) the XES signals with an O-*2p* character at higher excitation energies above 545 eV. These features are generally enhanced during the electrochemical charging. However, such enhancement is mostly from increased TM-O hybridization strength, which takes place in almost all battery electrodes and should not be counted as signatures of intrinsic oxygen redox reactions.

2.  A weak mRIXS feature with energy loss of about 2.4 eV (Feature-3) was observed in discharged electrodes. Its intensity also varied with electrochemical cycling, but was weakened in charged states. This stemmed from *d-d* excitations of the TM-*3d* states. The emergence of TM *d-d* features in O-*K* mRIXS is fundamentally interesting and deserves further theoretical studies. The reason behind the weakening during electrochemical charge remains unclear at this time, but was likely due to the overall broadening of spectroscopic features in a more covalent and amorphous phase at charged states.

3.  A sharp mRIXS feature at the 531.0 excitation and 523.7 eV emission energy (Feature-5) emerged when the oxygen redox reaction took place above 4.35 V during the initial charge. The emission energy of this feature matched the spectroscopic behavior of oxidized oxygen in reference compounds, such as $Li_2O_2$. More importantly, the intensity of this feature followed closely the oxygen redox behavior during the electrochemical charge and discharge cycling, providing a reliable signature to fingerprint the oxygen redox reactions in batteries.

4.  Additionally, close to the elastic line in mRIXS results, a weak feature with less than a 1 eV energy loss (Feature-4) behaved in the same way as feature-5 upon electrochemical cycling. This provides another weak, but clean, indicator of oxygen redox reactions, and suggests that the oxidized oxygen states in charged electrodes spontaneously trigger low-energy excitations in its molecular configurations, likely through electron-phonon (vibronic mode) coupling. Feature-4 thus not only senses the oxygen redox reactions in the system, but also provides an important hint on the molecular model of the oxidized oxygen states.

5.  As previously reviewed [8], mRIXS results simultaneously included various fluorescence yield channels through different intensity integrations along emission energy axis. Although not a focused topic in this article, it is important to note that integrating the intensity around the characteristic 531 eV emission energy opens up the opportunity for quantifying the oxygen redox evolution upon electrochemical cycling [17]. With enough signal statistic, the same approach

could be extended to the weak low-energy excitation feature-4, and could be employed to study almost all TM oxide based battery electrodes.

While the theoretical simulation for fundamentally understanding the oxygen redox features in O-*K* mRIXS remains very challenging, these experimental observations provide a critical benchmark on how to reliably detect and evaluate oxygen redox reactions in battery electrodes. Additionally, spectroscopic results will trigger extensive studies based on both material characterizations and theoretical calculations, which will eventually lead to technological developments and a fundamental understanding of the true mechanism of oxygen redox reactions in batteries.

**Author Contributions:** J.W., S.S., Q.L. and W.Y. organized and analyzed spectroscopic data; Q.L. and W.E.G. conducted the experiments; all authors discussed the results. J.W. and W.Y. wrote the paper with all authors reviewed and contributed to the manuscript.

**Funding:** This research was funded by DOE Office of Science User Facility, grant number DE-AC02-05CH11231.

**Acknowledgments:** This research used resources of the Advanced Light Source, which is a DOE Office of Science User Facility under contract no. DE-AC02-05CH11231. Q.L. acknowledges financial support from China Scholarship Council. Q.L. and S.Y. acknowledge financial support from 111 Project no. B13029. W.Y. acknowledges the support from the Energy Biosciences Institute through the EBI-Shell program.

**Conflicts of Interest:** The authors declare no conflict of interest.

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
