# Peer review of "Fingerprint Oxygen Redox Reactions in Batteries through High-Efficiency Mapping of Resonant Inelastic X-ray Scattering"

_condensedmatter, doi:10.3390/condmat4010005_

Round 1
Reviewer 1 Report
The manuscript submitted by Wu et al. demonstrates that high-efficiency full-range mapping of resonant inelastic X-ray scattering (mRIXS) is a powerful tool to resolve the O-K pre-edge features of NCM battery cathodes. Specifically, the authors claim that the two peaks at 531.0 eV and 523.7 eV excitation energy are signatures of the reversible redox reactions of the lattice oxygen in NCM cathodes.
This work is well designed, clearly articulated and reasonably discussed. The overall quality of this work definitely meets the publication requirement of the submitted journal. I have no major concerns but a few minor suggestions. The manuscript could be accepted after it is properly revised.
1. Please provide a schematic illustration showing the experimental set-up of mRIXS. A short description of how the set-up works is appreciated.
2. I assume that the conclusions obtained in this study only apply to NCM cathodes. If so, please state it explicitly in the manuscript.
3. Line 296: Please check if the "#" sign is a typo.
Author Response
Please see detailed response letter in PDF to address the minor technical comments from the reviewer.

Reviewer 2 Report
In this paper, the authors report on a comprehensive interpretation observed in O-K mRIXS of LR-NMC based battery cathodes involving oxygen redox reactions. For the latter, the authors have developed reliable probes through the full-range mapping of mRIXS. The subject is interesting and important from practical and theoretical point of view. The paper is well-organized, clearly structured, supported by literature, and the results have been discussed in a great detail. I have no particular suggestions for the authors, since the manuscript is error-free and the message is clear. I would only suggest the authors to enlarge the axes title of Figs 4 &5.
The manuscript indeed warrants the publication in Condensed Matter.
Author Response
Thank you and we optimized the figure display as the reviewer suggested.
Attached please see a response letter with the reviewer comments quoted for your convenience.
